# Electroredox carbene organocatalysis with iodide as promoter

Peng Zhou[1], Wenchang Li[1], Jianyong Lan[1] & Tingshun Zhu [1✉]

Oxidative carbene organocatalysis, inspired from Vitamin B1 catalyzed oxidative activation from pyruvate to acetyl coenzyme A, have been developed as a versatile synthetic method. To date, the α-, β-, γ-, δ- and carbonyl carbons of (unsaturated)aldehydes have been successfully activated via oxidative N-heterocyclic carbene (NHC) organocatalysis. In comparison with chemical redox or photoredox methods, electroredox methods, although widely used in mechanistic study, were much less developed in NHC catalyzed organic synthesis. Herein, an iodide promoted electroredox NHC organocatalysis system was developed. This system provided general solutions for electrochemical single-electron-transfer (SET) oxidation of Breslow intermediate towards versatile transformations. Radical clock experiment and cyclic voltammetry results suggested an anodic radical coupling pathway.

[1] School of chemistry, Sun Yat-sen University, Guangzhou 510006, China. ✉email: zhutshun@mail.sysu.edu.cn

Oxidative activation is a general activation mode which widely existed not only in the biological system, but also in synthetic chemistry. As an important transformation in mitochondria, oxidative activation from pyruvate to acetyl coenzyme A (CoA) have been discovered since 1937 (Fig. 1a)[1]. In this transformation, pyruvate was believed to be firstly reacted with thiamine pyrophosphate (TPP, Vitamin B1, VB1) to form a Breslow intermediate after decarboxylation, and followed by oxidation with pyruvate ferredoxin oxidoreductase (PFOR) (with two SET processes) towards an acyl azolium intermediate, which underwent thioesterification to give acetyl-CoA. In synthetic chemistry, besides 2-oxocarboxylic acids, aldehydes were also efficient substrate for this transformation. In 1968, Corey and coworkers developed an oxidative esterification of aldehydes using cyanide as catalyst and $MnO_2$ as oxidant[2]. Since 1977, different types of oxidative esterification transformations were developed, using thiazolium, imidazolium or triazolium NHC catalysts with various types of oxidants[3–11]. Among them, tetra-tert-butyl diphenoquinone (DQ), pioneered by Studer[9], was eventually developed as the most popular oxidant in NHC organocatalysis. After Studer's pioneer work of β-LUMO activation[12–15] with oxidative NHC organocatalysis in 2010, the remote activation modes involving α-[16–18], γ-[19–21], or δ-[22–24] carbon functionalization were rapidly developed in the later several years. Recently, SET redox activation mode was developed in NHC organocatalysis, with chemical redox[25–27] or photoredox methods[28–30].

With anodic electron transfer as green oxidant, electrochemical oxidation is one of the perfect choices in oxidation activation[31–40]. Inspired by the biomimic electrochemical oxidation of pyruvate[41,42], Boydston and coworkers[43,44] developed a pioneer work of electrochemical oxidative carbene organocatalysis in 2013. With thiazolium NHC catalyst, different aldehydes were smoothly oxidized to generate corresponding esters or thioesters in undivided cells with constant voltage direct current (DC). However, this system was limited to the (thio)esterification with thiazolium NHC catalyst, and with no further development in the later decade. The wonderful world of oxidative carbene organocatalysis[45–52] with different NHC catalysts (imidazolium, triazolium NHCs), different activation modes (α-, β-, γ-, or δ-carbon functionalization) and enantioselective transformations were still waiting for a general and efficient electrochemical oxidation system.

Inspired by the proposed concept of coupled electrolysis in Lin's work[53] in 2018, herein we developed a general electrochemical catalytic system for oxidative carbene organocatalysis. As shown in Fig. 1b, Breslow intermediate was anodic oxidized to radical cation intermediate[54,55] II (Fig. 1b, anodic event A), while iodine radical[56] was also generated on anode (Fig. 1b, anodic event B). The coupling of these two radicals gave intermediate III, which further affording acyl azolium intermediate IV after an eliminative regeneration of iodide ion. It is worth to note that iodine can poison carbene catalyst[57,58] and was never applied as oxidant in NHC organocatalysis. In our system, the concentration of NHC catalyst was much higher than that of radical intermediate II, however, with the help of non-uniform distribution of electrolysis system[59], iodine radical and ketyl radical II underwent radical coupling near the surface of anode before their dispersion into solvent system. The radical clock experiment of cis-2-phenylcyclopropane-1-carbaldehyde gave difference results with conventional chemical oxidation process. Radical intermediate in our system was believed to undergo a reversible ring-opening of cyclopropane, to give the trans- ester product (Fig. 1c).

In this work, a general solution was provided for electrochemical single-electron-transfer (SET) oxidation of Breslow intermediate towards versatile transformations.

## Results and discussion

**Reaction optimization.** We started our investigation by choosing the formal [4 + 2] annulation of enal 1a and hydrazone 2a as the model reaction[60,61] of oxidative γ-activation. In a constant current (1 mA) electrochemical system with platinum as both anode and cathode material, in the presence of $K_2CO_3$ and $n$-Bu$_4$NI in $CH_2Cl_2$, NHC precatalyst A[62] successfully catalyzed the reaction of enal 1a and hydrazone 2a, giving the desired product 3a in 79% isolated yield with 97% ee. Control experiments were conducted to indicate that both NHC. catalyst and electricity were essential for this reaction. (Table 1, entries 2 and 3). Using other solvents (such as DCE, THF and $CH_3CN$) instead of $CH_2Cl_2$ all led to decreased product yields (Table 1, entries 4–6). Using $n$-Bu$_4$NBF$_4$ instead of $n$-Bu$_4$NI as electrolyte was not viable, while using a mixture electrolyte with 20 mol% $n$-Bu$_4$NI and 80 mol% $n$-Bu$_4$NBF$_4$ gave 55% yield with 97% ee (Table 1, entry 8), indicating that direct anodic oxidation from Breslow intermediate (Fig. 1, intermediate I) to acyl azolium intermediate (Fig. 1, intermediate IV) was inefficient in this system. Iodide ion was needed as a promoter. Changing the anion of electrolyte to Br⁻ or changing the cation of electrolyte to Et$_4$N⁺ both afford lower yield (Table 1, entries 9–10). The effect of base was also investigated, $K_2CO_3$ showed better performance than other base such as $Cs_2CO_3$ or DBU (Table 1, entries 11–12). Platinum showed better performance than graphite as anode material (Table 1, entries 1 and 13).

## Substrate scope

**Substrate scope.** With optimized conditions in hand, the substrate scope of the model reaction was investigated (Fig. 2). Functional groups in the aromatic ring of the hydrazones such as

(a) The developing history of redox carbene organocatalysis

**Fig. 1 Electroredox carbene organocatalysis with iodide as promoter. a** The developing history of redox carbene organocatalysis. **b** This work: electroredox single-electron-transfer (SET) with NHC catalysis. **c** Different results of radical clock experiment with traditional DET models and our SET models.

**Table 1 Effect of Reaction Parameters[a].**

| Entry | variations | Yield (%)[b] | ee (%)[c] |
|---|---|---|---|
| 1 | None | 79 | 97 |
| 2 | No NHC **A** | 0 | – |
| 3 | No electricity | n.r. | – |
| 4 | DCE as solvent | 60 | 95 |
| 5 | THF as solvent | 64 | 96 |
| 6 | $CH_3CN$ as solvent | 30 | 93 |
| 7 | $n\text{-}Bu_4NBF_4$ as electrolyte | trace | – |
| 8 | 20 mol% $n\text{-}Bu_4NI$ + 80 mol% $n\text{-}Bu_4NBF_4$ as electrolyte | 55 | 97 |
| 9 | $n\text{-}Bu_4NBr$ as electrolyte | 30 | 95 |
| 10 | $Et_4NI$ as electrolyte | 68 | 95 |
| 11 | $Cs_2CO_3$ as base | 29 | 90 |
| 12 | DBU as base | 29 | 31 |
| 13 | Graphite as anode | 62 | 97 |

[a]Standard conditions: Pt anode, Pt cathode, **1a** (0.15 mmol), **2a** (0.1 mmol), NHC **A** (20%), $K_2CO_3$ (150%), $n\text{-}Bu_4NI$ (1.0 equiv.), $CH_2Cl_2$ (3 mL), at a constant current of 1.0 mA for 6 h (2.24 F·mol$^{-1}$) in IKA ElectraSyn 2.0 at room temperature.
[b]Yield of the isolated product.
[c]The enantiomeric ratio (ee) was determined by chiral stationary HPLC. *DCE* 1,2-dichloroethane, *DBU* 1,8-diazabicyclo[5.4.0]undec-7-ene.

fluoro, methoxy and bromo substituents worked well (**3a**–**3d**). 3-pyridyl and 2-furyl substituents in the hydrazone substrates were also tolerated (**3e** and **3f**). Various aryl substituents of the α,β-unsaturated aldehydes were all suitable for this transformation, giving the desired products in good yields with excellent ee (**3g**–**3j**). Substrates with different ester substituents were also tolerated, affording the corresponding products with good results (**3k** and **3 l**). To futher demonstrate the practicality of our synthetic methods, substrates derived from different bioactive moleculues were also tested. As shown in Fig. 2, lactames products derived from isoniazid (**3m**), probenecid (**3n**), febuxostat (**3o**), indometacin (**3q**) and dehydrocholic acid (**3q**) were all successfully obtained in moderate to good yield, with excellent ee.

Encouraged by the success of electrochemical oxidative reaction on enal γ-carbon, we next evaluated the electrochemical approach for the oxidative β-carbon reaction of enal. As exemplified in Fig. 3, in the model reaction[63] of formal [3 + 3] annulation of enals (**4**) and 1,3-dicarbonyls (**5**), both imidazolium catalyst **B** and triazolium catalyst **C** were applicable in our system (The optimized condition with two catalyst was slightly different in base and solvent, see Supplementary Information for details). Reactions of different 1,3-dicarbonyl compounds with cinnamaldehyde gave the lactone products effectively (**6a**–**6e**). Reactions of 2,4-pentanedione and enals with different aromatic substituents also successfully afford the corresponding products in moderate yield (**6f**–**6i**).

The electrocatalytic protocol for oxidative enal α-carbon atom functionalization was also studied. The formal [2 + 4] annulation of aliphatic aldehydes **7** and α,β-unsaturated ketones **8** was chosen as a model reaction[16,17] and the results were shown in Fig. 4 (see Supplementary Information for details of condition optimization). Different aliphatic aldehydes reacted with chalcone smoothly, and gave the lactone products in good yields with excellent enantioselectivities (**9a**–**9d**). Variation in the chalcone skeleton with different aromatic substituents had little influence on the of this reaction, and a broad range of groups, such as fluoro, chloro, methyl, methoxy and furyl groups were viable to get excellent ee (**9e**–**9i**).

For oxidative functionalization of aldehyde carbonyl carbon, an NHC-catalyzed dynamic kinetic resolution of hemiacetal was selected as a model reaction[64,65] and the results were shown in Fig. 5 (see Supplementary Information for details of condition optimization). One of the biggest challenges of this reaction was to prevent the anodic oxidation of hemiacetal **11** towards phthalic anhydride. Fortunately, our system was well kinetically controlled. The oxidation of Breslow intermediate was prior to that of hemiacetal **11**, no phthalic anhydride byproduct was observed. Different aromatic aldehydes were appliable in the reaction, giving chiral acetal product **12a**–**12d** with excellent ee.

**Miscellaneous reactions and gram-scale synthesis**. To further investigate the generality of our catalytic system, additional examples of different model reactions were tried and the results were summarized in Fig. 6. To our delight, the reaction system was quite general for different reactions, including δ-carbon[22] functionalization of enal **13** towards multisubstituted benzene **14** (Fig. 6a), β-carbon[66] functionalization of enal **4a** towards chiral lactam **16** (Fig. 6b) and γ-carbon[67] functionalization of enal **1a** towards multicyclic product **18** (Fig. 6c). Brief screening of solvent could find acceptable condition for these reactions (see Supplementary Information for details). The corresponding products were obtained in moderate yield with excellent ee. An initial test towards scale-up synthesis was also studied. As shown in Fig. 6d, the reaction of **1a** and **2a** on a 5 mmol scale underwent smoothly for 83.5 h, giving the desired product **3a** in 62% yield with 96% ee. In comparison, the traditional chemical oxidant strategy of this reaction may need at least 3 gram of oxidant **DQ** and will generate nearly the same amount of reductive byproduct (diphenyl diphenone). These results further demonstrate the generality and efficiency of the electrochemical oxidation system.

**Mechanistic studies**. Some controlled experiments were carried out for mechanistic study, the results were summarized in Fig. 7. In

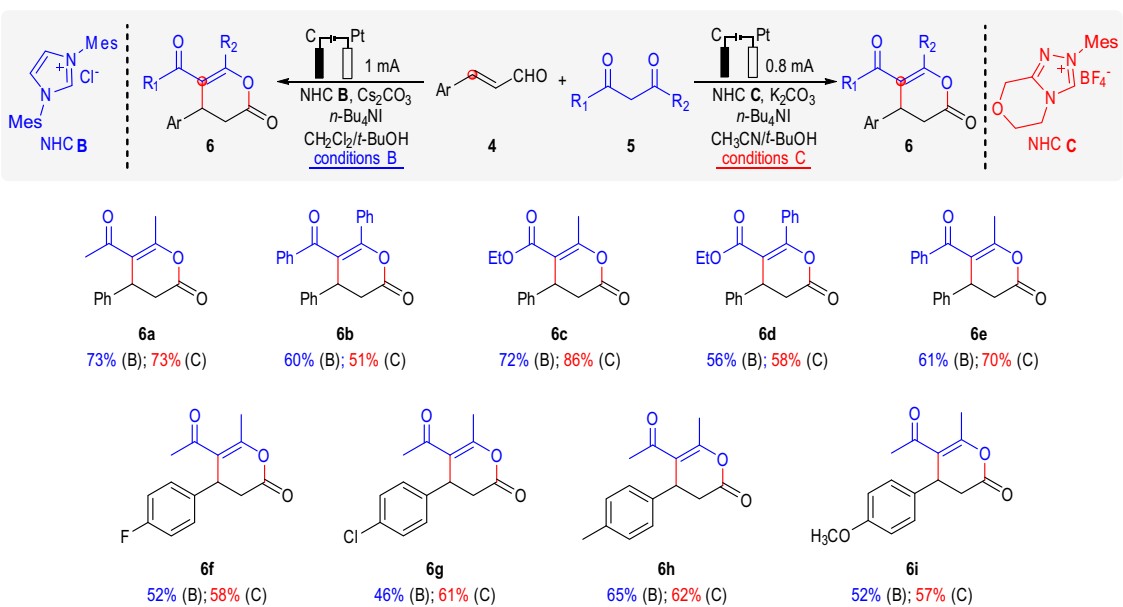

**Fig. 2 γ-carbon atom reaction: Scope of the reaction of enals with hydrazones.** Reaction conditions: Pt anode, Pt cathode, **1** (0.15 mmol), **2** (0.1 mmol), NHC **A** (20%), K$_2$CO$_3$ (150%), *n*-Bu$_4$NI (1.0 equiv.), CH$_2$Cl$_2$ (3 mL), at a constant current of 1 mA for 6 h (2.24 F·mol$^{-1}$) in IKA ElectraSyn 2.0 at room temperature. The ee was determined by chiral stationary HPLC. $^a$ THF/CH$_2$Cl$_2$ (2 mL/1 mL) used as solvent.

**Fig. 3 β-carbon atom reaction: Scope of the reaction of enals with 1,3-dicarbonyls.** Conditions B: graphite anode, Pt cathode, **4** (0.1 mmol), **5** (0.2 mmol), NHC **B** (30%), Cs$_2$CO$_3$ (30%), *n*-Bu$_4$NI (1.0 equiv.), CH$_2$Cl$_2$/*t*-BuOH (2 mL/ 1 mL), at a constant current of 1 mA for 6 h (2.24 F·mol$^{-1}$) in IKA ElectraSyn 2.0 at room temperature. Conditions C: graphite anode, Pt cathode, **4** (0.1 mmol), **5** (0.2 mmol), NHC **C** (20%), K$_2$CO$_3$ (20%), *n*-Bu$_4$NI (1.0 equiv.), CH$_3$CN/*t*-BuOH (1.5 mL/ 1.5 mL), at a constant current of 0.8 mA for 10 h (2.98 F·mol$^{-1}$) in IKA ElectraSyn 2.0 at room temperature.

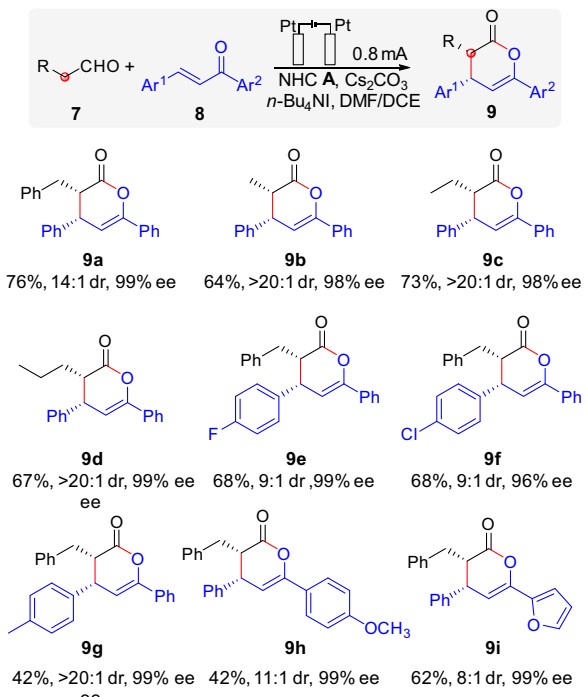

**Fig. 4 α-carbon atom reaction: Scope of the reaction of aldehydes with chalcone enones.** Reaction conditions: Pt anode, Pt cathode, **7** (0.25 mmol), **8** (0.1 mmol), NHC **A** (30%), $Cs_2CO_3$ (30%), $n$-Bu₄NI (1.0 equiv.), DMF/DCE (2 mL/ 1 mL), at a constant current of 0.8 mA for 9 h (2.68 F·mol⁻¹) in IKA ElectraSyn 2.0 at room temperature. The diastereomeric ratio (dr) was determined by ¹H NMR analysis of the crude products. The ee was determined by chiral stationary HPLC.

**Fig. 5 The enantioselective of hydroxyphthalide acylation by carbene-catalyzed dynamic kinetic resolution.** Reaction conditions: Pt anode, Pt cathode, **10** (0.18 mmol), **11** (0.1 mmol), NHC **A** (20%), DIEA (100%), $n$-Bu₄NI (1.0 equiv.), THF (3 mL), at a constant current of 1 mA for 6 h (2.24 F·mol⁻¹) in IKA ElectraSyn 2.0 at room temperature. The ee was determined by chiral stationary HPLC.

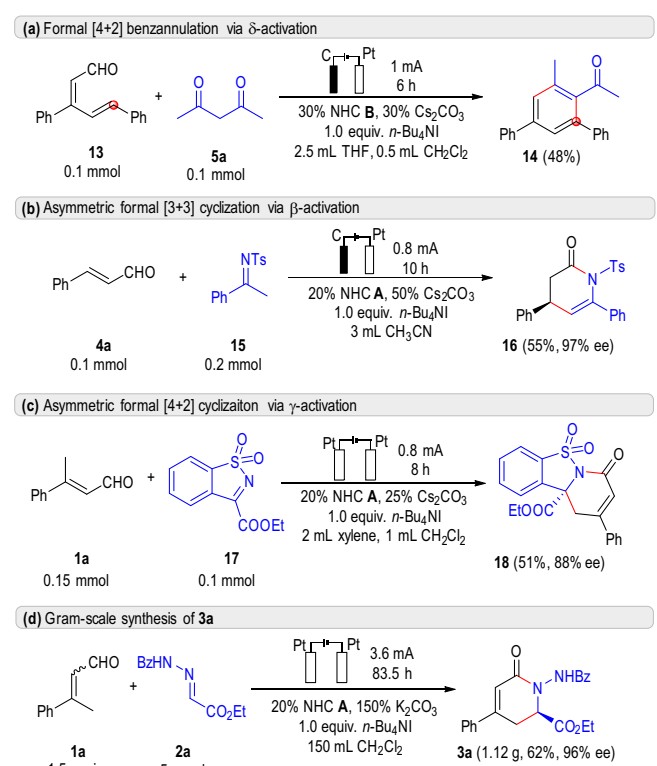

**Fig. 6 Miscellaneous reactions and gram-scale synthesis. a** Formal [4 + 2] benzannulation via δ-activation. **b** Asymmetric formal [3 + 3] cyclization via β-activation. **c** Asymmetric formal [4 + 2] cyclization via γ-activation. **d** Gram-scale synthesis of **3a**.

showed no activity as we expected (Fig. 7d). Another possible pathway involving iodination of enal substrate was also excluded by the iodination control test (Fig. 7e). To further confirm the existence of the NHC-attached ketyl radical intermediate (Fig. 1, intermediate **II**), a radical clock experiment was carried out (Fig. 7f). Aldehyde *cis*-**20** with cyclopropyl group was conducted in our reaction system to undergo an oxidative esterification reaction. As we expected, due to the radical isomerization[68,69] towards a more thermal dynamically stabled *trans*- structure, ester product *trans*-**21** was obtained as main product. To exclude the possibility of α-racemization of aldehyde under basic condition, *cis*-**20** was put under standard condition, and no isomerized *trans*-**20** was observed in crude NMR or HPLC. In comparison, conventional chemical oxidation process, no matter with TBAI or not, only gave non-isomerized ester *cis*-**21**. These results supported our mechanistic proposal of anodic SET oxidation towards radical intermediate (Fig. 1, intermediate **II**).

The electrochemical properties of different reactants and reagents were investigated in cyclic voltammetry experiments (Fig. 8). Direct electrochemical oxidation of enal **1a** required high potential (>2 V vs. SCE, see Supplementary Information for details). After the addition of NHC **A** and DBU, the formation of Breslow intermediate lead to a dramatically decline in oxidation potential to below −1.0 V (Fig. 8, blue line)[70] Mixing Breslow intermediate and TBAI only lead to slightly shift with Breslow intermediate's CV signals (Fig. 8, blue line V.S. red line), but obvious changes involving TBAI's (Fig. 8, brown line V.S. red line). The second oxidation involving iodine radical to iodine cation and both the back-reduction were all disappeared, which was expected from the irreversible radical coupling between iodine radical and NHC ketyl radical **II**. The lower oxidation potential of Breslow intermediate may also ensure the sufficient

the presence of 50 mol% I₂, the [4 + 2] annulation of enal **1a** and hydrazone **2a** was fully suppressed (Fig. 7a). When 10 mol% I₂ was subjected to the optimized reaction conditions (20 mol% NHC), half of the NHC catalyst **A** were believed to be poisoned and the result was nearly the same with the reaction with 10% NHC catalyst **A** (Fig. 7b). This result told us that the poisoned catalyst was probably inert in the reaction. Without substrates, NHC catalyst and electrolyte $n$-Bu₄NI was not compatible in the electrochemical system. All of the NHC catalysts were consumed and about 50% yield of iodination product **D** or **E** was isolated, (Fig. 7c), and

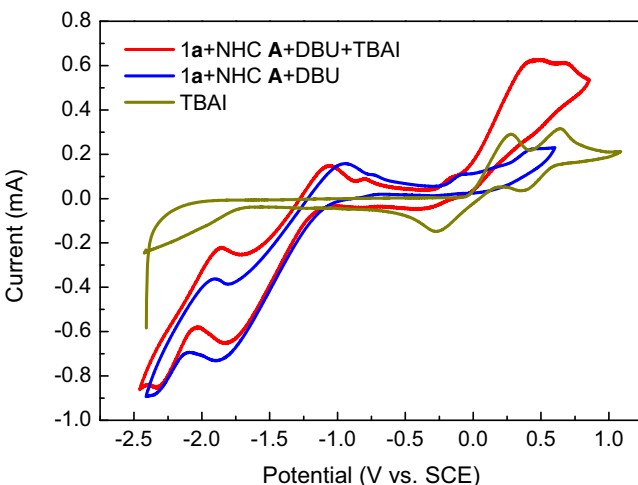

**Fig. 7 Mechanistic studies. a** Control experiment with $I_2$ as chemical oxidant. **b** Poison effect of the $I_2$ to NHC catalyst. **c** poison effect: iodination of NHC catalyst under the electrochemical conditions. **d** Reversibility test of poison effect. **e** the possibility of γ-iodination pathway. **f** Radical clock experiments.

**Fig. 8 Cyclic voltammograms.** General condition: solvent $CH_2Cl_2$ with 0.1 M $n$-Bu$_4$NBF$_4$ as supporting electrolyte; brown line: 0.1 mmol $n$-Bu$_4$NI; blue line: **1a** (0.3 mmol), NHC **A** (0.15 mmol, 50%), and DBU (0.15 mmol, 50%); red line: **1a** (0.3 mmol), NHC **A** (0.15 mmol, 50%), DBU (0.15 mmol, 50%) and $n$-Bu$_4$NI (0.1 mmol).

capture of iodine radical and prevent its catalyst poisoning effect. Cyclic voltammetry experiments involving other model reactions were also carried out, and all of them gave similar results (see Supplementary Information for details).

In this work, we have developed a modular method of anodic coupled electrolysis system in which NHC catalysis is merged with cooperative iodide ion and electrocatalysis. This coupled electrolysis system avoids the usage of big amount of chemical oxidant in oxidative NHC organocatalysis. The green reaction system is readily available for different activation modes (α-, β-, γ-, δ- or carbonyl carbon functionalization), different reaction types (cyclization, benzannulation, dynamic kinetic resolution, etc.) and scale up productions. Mechanism studies involving controlled test, radical clock experiments and cyclic voltammetry measurements provided sufficient evidence to support our proposal of anodic oxidation induced radical coupling, which differs with conventional chemical oxidative approach. Learning from the wonderful features of oxidative NHC organocatalysis (two-electron oxidation or SET oxidation), we believe the electrosynthesis method not only provides more feasibility for large-scale applications, but also opens new avenues in the area of NHC-catalyzed radical reactions. Further studies regarding electroredox activation of deoxy-Breslow intermediate and other NHC attached intermediates are in progress in our laboratory.

## Methods

**General produce for electrochemical [4 + 2] annulation of 1 with 2 by the catalysis of NHC A**. The ElectraSyn vial (5 mL) with a stir bar was charged with β-methyl enals 1 (0.15 mmol, 1.5 equiv.), NHC A (0.02 mmol, 20%), $K_2CO_3$ (0.15 mmol, 1.5 equiv.), $n$-Bu$_4$NI (0.1 mmol, 1.0 equiv.) and hydrazones 2 (0.1 mmol, 1.0 equiv.) followed by anhydrous $CH_2Cl_2$ (3.0 mL). The ElectraSyn vial cap equipped with anode (Pt) and cathode (Pt) were inserted into the mixture. After pre-stirring for 2 min, the Electrasyn vial was connected to the Electrasyn 2.0 and the reaction mixture was electrolyzed under a constant current of 1.0 mA for a total reaction time of 6 h accompanied by magnetic stirring. The ElectraSyn vial cap was removed, and electrodes were rinsed with EtOAc, which was combined with the crude mixture. After concentrated under reduced pressure, the crude residue was purified via flash column chromatography to afford the desired product 3

**General procedure for electrochemical [3 + 3] annulation of 4 with 5 by the catalysis of NHC B**. The ElectraSyn vial (5 mL) with a stir bar was charged with α,β-unsaturated aldehydes 4 (0.1 mmol, 1.0 equiv.), NHC B (0.03 mmol, 30%), $Cs_2CO_3$ (0.03 mmol, 30%.), $n$-Bu$_4$NI (0.1 mmol, 1.0 equiv.) and 1,3-dicarbonyl derivatives 5 (0.2 mmol, 2.0 equiv.) followed by anhydrous $CH_2Cl_2$ (2.0 mL) and $t$-BuOH (1.0 mL). The ElectraSyn vial cap equipped with anode (graphite) and cathode (Pt) were inserted into the mixture. After pre-stirring for 2 min, the Electrasyn vial was connected to the Electrasyn 2.0 and the reaction mixture was electrolyzed under a constant current of 1.0 mA for a total reaction time of 6 h accompanied by magnetic stirring. The ElectraSyn vial cap was removed, and electrodes were rinsed with EtOAc, which was combined with the crude mixture. After concentrated under reduced pressure, the crude residue was purified via flash column chromatography to afford the desired product 6.

**General procedure for electrochemical [3 + 3] annulation of 4 with 5 by the catalysis of NHC C**. The ElectraSyn vial (5 mL) with a stir bar was charged with α,β-unsaturated aldehydes 4 (0.1 mmol, 1.0 equiv.), NHC C (0.02 mmol, 20%), $K_2CO_3$ (0.02 mmol, 20%.), $n$-Bu$_4$NI (0.1 mmol, 1.0 equiv.) and 1,3-dicarbonyl derivatives 5 (0.2 mmol, 2.0 equiv.) followed by anhydrous $CH_3CN$ (1.5 mL) and $t$-BuOH (1.5 mL). The ElectraSyn vial cap equipped with anode (graphite) and cathode (Pt) were inserted into the mixture. After pre-stirring for 2 min, the Electrasyn vial was connected to the Electrasyn 2.0 and the reaction mixture was electrolyzed under a constant current of 0.8 mA for a total reaction time of 10 h accompanied by magnetic stirring. The ElectraSyn vial cap was removed, and electrodes were rinsed with EtOAc, which was combined with the crude mixture. After concentrated under reduced pressure, the crude residue was purified via flash column chromatography to afford the desired product 6.

**General produce for electrochemical [2 + 4] annulation of 7 with 8 by the catalysis of NHC A**. The ElectraSyn vial (5 mL) with a stir bar was charged with enones 8 (0.1 mmol, 1.0 equiv.), NHC A (0.03 mmol, 30%), $Cs_2CO_3$ (0.03 mmol, 30%), $n$-Bu$_4$NI (0.1 mmol, 1.0 equiv.) and aldehydes 7 (0.25 mmol, 2.5 equiv.) followed by anhydrous DMF (2.0 mL) and DCE (1.0 mL). The ElectraSyn vial cap equipped with anode (Pt) and cathode (Pt) were inserted into the mixture. After pre-stirring for 2 min, the Electrasyn vial was connected to the Electrasyn 2.0 and the reaction mixture was electrolyzed under a constant current of 0.8 mA for a total reaction time of 9 h accompanied by magnetic stirring. The ElectraSyn vial cap was removed, and electrodes were rinsed with EtOAc, which was combined with the crude mixture, and extracted with $H_2O$ three times. After concentrated the organic phase under reduced pressure, the crude residue was purified via flash column chromatography to afford the desired product 9.

**General produce for electrochemical asymmetric acylation of hydroxyphthalide by carbene-catalyzed dynamic kinetic resolution**. The ElectraSyn vial (5 mL) with a stir bar was charged with aldehydes 10 (0.18 mmol, 1.8 equiv.), NHC A (0.02 mmol, 20%), DIEA (0.1 mmol, 100%.), $n$-Bu$_4$NI (0.1 mmol, 1.0 equiv.) and hydroxyphthalide 11 (0.1 mmol, 1.0 equiv.) followed by anhydrous THF (3.0 mL). The ElectraSyn vial cap equipped with anode (Pt) and cathode (Pt) were inserted into the mixture. After pre-stirring for 2 min, the Electrasyn vial was connected to the Electrasyn 2.0 and the reaction mixture was electrolyzed under a constant current of 1 mA for a total reaction time of 6 h accompanied by magnetic stirring. The ElectraSyn vial cap was removed, and electrodes were rinsed with EtOAc, which was combined with the crude mixture. After concentrated under reduced pressure, the crude residue was purified via flash column chromatography to afford the desired product 12.

## Data availability

Supplementary information is available in the online version of the paper. Data supporting the results of this work are available within this paper or its Supplementary Information, and are also available upon request from the corresponding author.

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

## Acknowledgements

Generous financial support for this work is provided by National Natural Science Foundation of China (22071269), the Pearl River Recruitment Program of Talent (2019QN01L149).

## Author contributions

P.Z. conducted most of the experiments; P.Z. and W.L. prepared substrates for the reaction scope evaluation; P.Z. and J.L. analyzed the data. T.Z. conceptualized and directed the project, and drafted the manuscript with the assistance from all co-authors. All authors contributed to discussions.

## Competing interests

The authors declare no competing interests
