## [Peer Review File · Nature Communications]

REVIEWER COMMENTS

Reviewer #1 (Remarks to the Author):

This manuscript describes an electroredox protocol to convert the NHC Breslow intermediate to acyl azolium using iodide as a promoter. The authors showed that concurrent single-electron oxidation occurred on the anode to yield the corresponding ketyl radical and iodine radical. Radical-radical combination, followed by elimination, yielded the acyl azolium species that can react with various partners. This study tested several annulation reactions that were reported using external oxidants and the results were quite good. Radical clock experiments confirmed the radical nature of the redox process, which is in contrast to the two-electron mechanism using DQ. In general, this is a simple approach to the generation of NHC ketyl radicals without the need for external oxidants. Publication may be considered.

Technical issues

1. The title claims "iodine cocatalyst". However, all reactions were carried out using 1 eq. nBu₄I. Technically, iodide was not used in a catalytic amount and should not be called a cocatalyst. "iodine cocatalyst" should be an iodide promoter.
2. In the background discussion, the authors mentioned that iodine reacts NHC to "poison" the catalyst. In fact, adding I₂ shut down the electroredox reaction as well. In their reactions, iodide was used and no iodine (I₂) was generated in the catalytic cycle (although I cannot rule out the possibility of some I₂ being formed as an off-cycle side product). So, it does not make much sense to discuss this "poisonous effect", especially claiming this work overcomes this problem. In the abstract, it was stated "With the help of non-uniform distribution of electrolysis system, NHC and iodine, which was normally not compatible in chemical reaction, cooperated well in the electrochemical system". There is little evidence that I₂ was compatible with their reactions. The abstract and related discussions should be revised.
3. It is peculiar that the oxidation potentials of Breslow intermediate and iodide are similar. Refer to a recent paper by Martin (*Angew. Chem. Int. Ed.* 2021, 60, 26783).
4. A control experiment is missing. Does treatment of cis-20 with NHC, Cs₂CO₃, DQ, nBu₄NI, and MeOH lead to the corresponding trans-ester 21?

Reviewer #2 (Remarks to the Author):

In this manuscript, Zhu and coworkers report the NHC-catalyzed radical reactions under electrochemical conditions with iodine as a co-catalyst. The method avoids the need for stoichiometric chemical oxidants and generates an important class of acyl azolium intermediates, which can be applied to various enantioselective transformation reactions such as cyclization, benzannulation, dynamic kinetic resolution etc.

Therefore, I recommend this paper be published in the journal after the following points are addressed. Before its acceptance, however, I believe that a couple minor revisions are warranted.

1. Hydrazones are good substrates for the formal [4+2] annulation with enal, but I was wondering whether oximes or imines can be tolerated in this protocol?
2. In order to further demonstrate the practicality of the synthetic method, I suggest authors include several bioactive molecules in the scope or a synthetic application if possible.
3. Compared to previous reported methods (Ref. 40, 41), iodide plays an vital role in this reaction, Why intermediate I is not oxidized to give acyl azolium through direct anodic oxidation?
4. In comparison, conventional chemical oxidation process only gave non-isomerized ester cis-21, but ester product trans-21 was obtained as main product in electrochemical reaction system, why?

Reviewer #3 (Remarks to the Author):

Zhu and coworkers reported a series of electrochemical reactions based on NHC and iodine cocatalyst system. Instead of adding external chemical oxidant, anodic oxidation was applied to this catalyst system. It is noted that the racemic version was already reported by Chi using polyhalides as oxidants (*Angew. Chem. Int. Ed.* 2017, 56, 2942–2946), but this work was not mentioned in the main text or reference. By switching to the chiral NHC catalysts, the authors could achieve high enantioselectivities. However, the similar transformation of hydroxyphthalide acylation (Figure 5) was also reported (*Nat Commun.* 2019, 10, 1675), the authors failed to cite the related work.

Although the realizations of such chiral transformations are important progress, anodic oxidation did not show any different reactivity compared to the racemic version. Considering these imidazolium and triazolium carbene catalysts are widely used in this chemistry, the authors did not clearly delineate why this study is different from others in the literature. As stated above, it would be more

suitable for a specialty journal with a high organic chemistry readership, than for a broad chemistry audience.

Referee #1 was supportive for our proposal of NHC ketyl radicals, and raised valuable suggestions involving mechanism studies.

1. In general, this is a simple approach to the generation of NHC ketyl radicals without the need for external oxidants.

Response: the development of electrochemical generation of NHC ketyl radicals in our manuscript is valuable not only because of non-requirement for external oxidants, but also, more importantly, more flexible and controllable oxidation mode and more solid evidence to support the existence of NHC ketyl radicals. As shown in the following figure, in the precedented cases of NHC ketyl radicals with chemical oxidants, due to the strict redox ability requirement of Breslow intermediate and reactant (RX, as oxidant at the same time) the selection of substrates was limited. In our system, with the help of the controllable redox ability of electrochemical system, aromatic/aliphatic aldehydes and enals are all applicable for different reactions. It is also worth to note that the radical clock experiment in our system is the first successful one to directly prove the existence of NHC ketyl radical. In previous examples, radical clock experiments with other radical reaction partner were shown as indirect evidence (related examples were added as ref 26-27).

2. The title claims “iodine cocatalyst”. However, all reactions were carried out using 1 eq. *n*Bu₄I. Technically, iodide was not used in a catalytic amount and should not be called a cocatalyst. “iodine cocatalyst” should be an iodide promoter.

Response: control test with 20 mol% *n*-Bu₄I (TBAI) was added in Table 1, entry 8, and related description was added in the manuscript “Using *n*-Bu₄NBF₄ instead of *n*-Bu₄NI as electrolyte was not viable, while using a mixture electrolyte with 20 mol% *n*-Bu₄NI and 80 mol% *n*-Bu₄NBF₄ gave 55% yield with 97% ee, indicating that direct anodic oxidation from Breslow intermediate (Fig. 1, intermediate I) to acyl azolium intermediate (Fig. 1, intermediate IV) was inefficient in this system. Iodine anion was needed as a cocatalyst.”

3. In the background discussion, the authors mentioned that iodine reacts NHC to “poison” the catalyst. In fact, adding I₂ shut down the electroredox reaction as well. In their reactions, iodide was used and no iodine (I₂) was generated in the catalytic cycle (although I cannot rule out the possibility of some I₂ being formed as an off-cycle side product). So, it does not make much sense to discuss this “poisonous effect”, especially claiming this work overcomes this problem. In the abstract, it was stated “With the help of non-uniform distribution of electrolysis system, NHC and iodine, which was normally not compatible in chemical reaction, cooperated well in the

electrochemical system” . There is little evidence that I2 was compatible with their reactions. The abstract and related discussions should be revised.

Response: to clarify the catalyst poison effect, a control experiment was added as Fig. 7c to show the incompatibility of NHC catalyst and *n*-Bu₄NI in the electrochemical system. Indeed, the formation of Breslow intermediate is reversible, and dimerization of iodine radical to form I₂ is very fast. If the solvent system is uniform, the concentration of NHC catalyst may be much higher than that of NHC ketyl radical, then the iodine radical or I₂ may possibly react with NHC catalyst to cause catalyst poisoning. In our system, both iodine radical and the NHC ketyl radical were generated on the surface of anode. As a result, they were likely to react with each other before dispersion into the solvent system.

Related descriptions were added in the main-text as “Without substrates, NHC catalyst and electrolyte *n*-Bu₄NI was not compatible in the electrochemical system. All of the NHC catalysts were consumed and about 50% yield of iodination product **D** or **E** was isolated, (Fig. 7c), which showed no activity as we expected (Fig. 7d).”

The descriptions of the mechanism in the third paragraph were also revised: “In our system, the concentration of NHC catalyst was much higher than that of radical intermediate **II**, however, with the help of non-uniform distribution of electrolysis system, iodine radical and ketyl radical **II** underwent radical coupling near the surface of anode before their dispersion into solvent system.”

c) Poison effect: iodination of NHC catalyst under the electrochemical conditions

4. It is peculiar that the oxidation potentials of Breslow intermediate and iodide are similar. Refer to a recent paper by Martin (Angew. Chem. Int. Ed. 2021, 60, 26783).

Response: CV test with extending scanning range was carried out, and redox potential signals below – 1.0 V was found. We try our best to purify all of the solvents and reagents, and confirm the redox potential signals of 0.23 V (similar with iodine as previous mentioned) were intrinsic feature of the Breslow intermediate rather than impurity effect. We guess it may be due to the different redox potential of OH-type and O-type of Breslow intermediate, or some other unexpected effect.

All of the descriptions involving “oxidation potentials of Breslow intermediate and iodide are similar” was revised. All of the CV figure in SI were changed to an extended scanning range version.

Figure 8 was changed (as shown below) to show the obvious differences of iodine anion's CV signals in the reaction system. The second oxidation peak and all the reduction peaks were disappeared, indicating that the iodine radical rapidly and irreversibly reacted with NHC ketyl radical. Related description was revised “Mixing Breslow intermediate and TBAI only lead to a slight shift with Breslow intermediate's CV signals (Fig. 8, blue line V.S. red line), but obvious changes involving TBAI's (Fig. 8, brown line V.S. red line). The second oxidation involving iodine radical to iodine cation and both the back-reduction were all disappeared, which was expected from the irreversible radical coupling between iodine radical and NHC ketyl radical **II**.”

Fig. 8 | Cyclic voltammograms recorded in CH₂Cl₂

5. A control experiment is missing. Does treatment of cis-20 with NHC, Cs₂CO₃, DQ, nBu₄NI, and MeOH lead to the corresponding trans-ester 21?

Response: the control experiment is added in Fig. 7, related description was added as “In comparison, conventional chemical oxidation process, no matter with TBAI or not, only gave non-isomerized ester cis-21.”. And the purpose of the control test in Fig. 7f were also added as “exclude the possibility of epimerization”, “different products with two-electron-mechanism using DQ”, for better understanding.

Referee #2 was supportive for acceptance and suggested some minor revisions.

1. Hydrazones are good substrates for the formal [4+2] annulation with enal, but I was wondering whether oximes or imines can be tolerated in this protocol?

Response: as shown below, two examples were tried. But unfortunately, the reactions involving imines and oximes remained unsuccessful.

2. In order to further demonstrate the practicality of the synthetic method, I suggest authors include several bioactive molecules in the scope or a synthetic application if possible.

Response: as shown below, examples derived from isoniazid, probenecid, febusostat, indomethacin or dehydrocholic acid were added in Fig. 2 as product 3m-3q. Related description were added as “To further demonstrate the practicality of our synthetic methods, substrates derived from different bioactive molecules were also tested. As shown in Fig.2, lactams products derived from isoniazid (3m), probenecid (3n), febusostat (3o), indometacin (3q) and dehydrocholic acid (3q) were all successfully obtained in moderate to good yield, with excellent ee.”

3. Compared to previous reported methods (Ref. 40, 41), iodide plays an vital role in this reaction, Why intermediate I is not oxidized to give acyl azolium through direct anodic oxidation?

Response: as mentioned in ref 70, unlike the thiazolium attached radical intermediate, imidazolium or triazolium attached ketyl radical **II** was unstable, and possibly cause the low efficiency of direct anodic oxidation to give acyl azolium. In all cases of our reactions, iodine anion cocatalyst were necessary to ensure acceptable yields.

4. In comparison, conventional chemical oxidation process only gave non-isomerized ester cis-21, but ester product trans-21 was obtained as main product in electrochemical reaction system, why?

Response: in the electrochemical reaction system, SET process was involved, the NHC ketyl radical attached to a cyclopropane was likely to undergo reversible ring opening/closing, which cause the epimerization. In the conventional chemical oxidation process, Breslow intermediate was directly oxidized to acyl azolium via double-electron-transfer process (the radical intermediate was not involved), so no epimerization occurred. Two reference (ref 68-69) were added. Similar radical clock experiments were carried out to prove the existence of radicals in these references.

Referee #3 had some questions about the differences between electro-oxidation method and the preceded chemical oxidation one, and raised valuable suggestions involving refences citing.

1. It is noted that the racemic vision was already reported by Chi using polyhalides as oxidants (Angew. Chem. Int. Ed. 2017, 56, 2942 – 2946), but this work was not mentioned in the main text or reference. By switching to the chiral NHC catalysts, the authors could achieve high enantioselectivities. However, the similar transformation of hydroxyphthalide acylation (Figure 5) was also reported (Nat Commun. 2019, 10, 1675), the authors failed to cite the related work

Response: the two references were added as ref-11 and ref-65

2. Considering these imidazolium and triazolium carbene catalysts are widely used in this chemistry, the authors did not clearly delineate why this study is different from others in the literature.

Response: the mechanism of our methods is completely different from the widely reported oxidative NHC organocatalysis, which was obviously shown in the comparison of radical clock experiment as shown below (also in Fig. 7f). In the electrochemical reaction system, single-electron-transfer (SET) process was involved, Breslow intermediate was oxidized to ketyl radical intermediate which was likely to undergo reversible cyclopropane ring opening/closing, and cause epimerization, *trans*-ester product was obtained. In the conventional chemical oxidation process, Breslow intermediate was directly oxidized to acyl azolium via double-electron-transfer (DET) process (the radical intermediate was not involved), so no epimerization occurred, *cis*-ester product was obtained.

Two reference (ref 68-69) were added. Similar radical clock experiments were carried out to prove the existence of radicals in these references. The purpose of the control test and radical clock experiment were also added in Fig. 7f to make it easier understood.

6 additional examples involving bioactive molecules derivatives were also added to show the practicality of our methods. (Fig. 2, product **3m-3q**)

We hope the above revision can meet the requirements for publication. Thank you very much again for your assistance with our manuscript.

Sincerely,
Tingshun Zhu

REVIEWER COMMENTS

Reviewer #1 (Remarks to the Author):

Responses to justify iodine as cocatalyst were not convincing. Although Table 1, entry 8 showed 20 mol% n-Bu₄NI, in combination with 80 mol% n-Bu₄NBF₄, gave 55% yield (2.75 turnover number of iodide, hardly catalytic), substrate tables were all generated using 1.0 eq. iodide. Therefore, "iodine as cocatalyst" shall be removed from the title and related discussions. Related to this topic, iodide was used, not iodine. The poisoning effect on NHC was primarily from I₂, not iodide. As the authors showed in Fig 7f, the use of stoichiometric iodide was not poisonous to the generation of acylazolium. There is no need to emphasize iodine poisoning. Fig. 1c should be removed as this manuscript did not address this issue.

Tingshun Zhu
zhutshun@mail.sysu.edu.cn
Professor, School of Chemistry
Sun Yat-Sen University, Guangzhou, China

May 25, 2022

Manuscript number: NCOMMS-21-46442A-Z

Referee #1 raised valuable suggestions involving the description of “cocatalyst” and “poisoning effect”.

1. Responses to justify iodine as cocatalyst were not convincing. Although Table 1, entry 8 showed 20 mol% n-Bu₄NI, in combination with 80 mol% n-Bu₄NBF₄, gave 55% yield (2.75 turnover number of iodide, hardly catalytic), substrate tables were all generated using 1.0 eq. iodide. Therefore, "iodine as cocatalyst" shall be removed from the title and related discussions. Related to this topic, iodide was used, not iodine. The poisoning effect on NHC was primarily from I₂, not iodide. As the authors showed in Fig 7f, the use of stoichiometric iodide was not poisonous to the generation of acyl azolium. There is no need to emphasize iodine poisoning. Fig. 1c should be removed as this manuscript did not address this issue.

Response:

- the title was changed to “Electroredox Carbene Organocatalysis with Iodide as Promoter”.
- All description of “cocatalyst” was revised to “promoter”.
- Figure 1c was removed
- “iodine anion” was revised to “iodide ion”
- Descriptions to emphasize iodine poisoning was removed.

We really appreciate the referee suggestions, and have revised the manuscript accordingly. Main changes in the revised manuscript were with “track changes” mode.